# Contribution of peri-urban land use and agriculture to entropy and food of mega-cities: On sustainability, planning by control theory and recycling of organics

Ernst-August Nuppenau ⓘ *

Justus-Liebig-University, Giessen, Germany

* eanuppenau@gmail.com

## Abstract

In this article we propose an ecological economics orientation of peri-urban development, looking at land use planning, local food and entropy. Based on a mathematical model, we present an operational concept of minimizing negative externalities within a given population. The model applies control theory. The concept is introduced to facilitate closing cycles, conduct spatial planning and reduce costs to achieve the ecological target of improved entropy. To this end, we look at more soundly defined metropolitan areas. An emphasis is placed on optimally assuring space for urban agriculture and on enabling recycling in ever-growing cities. Our concept is grounded in the use of peri-urban agriculture and regional food provision as an integrated system, which is based on the recycling of organic matter. Firstly, we reference current unhealthy developFfigments and show how cycles were removed as growth occurred. Secondly, as market-oriented city expansions showed limited scope for peri-urban farming, we suggest entering into regional planning. Planning shall ensure a better use of space and can be based on organic matter recycling (composting, slurry, etc.). The article provides a theoretical background for the occurrence of modified land use (systems). These systems shall alleviate some external burdens of large and growing cities. The approach looks at ecological, economic and social aspects in parallel, to outline principles for more sustainable land use, including peri-urban land. Methodologically we offer land use modelling, looking at interactions of industry, residence, and farming. The functions of a city are integrated into a methodical approach of distance from centre to periphery.

## 1 Introduction

In the current face of globalisation and further expected urban growth, peri-urban agriculture and recycling of nutrients is an undervalued issue [1]. To assist rising cities in limiting their negative ecological impact, we suggest spatial planning. With this in mind, we will work on the entropy of cities [2] (clarified in detail later) as a goal for planning. A fundamental and simple

**Data Availability Statement:** no emprical data included. It is a theoretical article.

**Competing interests:** The authors have declared that no competing interests exist.

prerequisite for sustainable development is that cities of the future must control their metabolisms again [2]. In future, planning and management of urban areas should embody a stewardship of the land resources needed to absorb and transform urban organic wastes. Based on planning and a waste management system, a regional outlay of city functions is needed involving peri-urban agriculture as a key function.

For instance, although wastewater is an important source of emissions to water-bodies and comprises industrial effluents and household connections, it can be used in agriculture [3], if properly dealt with. Yet, in peri-urban areas it is mostly land which can be used to recycle that is missing or scarce. If one aims for the alleviation of ecological problems in the nexus of urban to peri-urban and peri-urban to rural areas, it may be wise to anticipate land use planning for more peri-urban agriculture. Then one can discuss ecological, social and economic effects systematically framing urban functions spatially. We will work on a spatial outline of residence, industry and peri-urban farming.

This paper will discuss how to develop strategies that bring about a more sustainable pattern of urban growth by way of introducing a viable peri-urban interface. In the context of an envisaged further rapid urban growth of cities, industrialisation (income) and rural-urban migration (residence), we, particularly suggest a redirection and new types of schemes for peri-urban land use systems. Secondly and in respect thereof, we deliver a problem statement, show how complexity can be reduced, and finally sketch a planning modelling which applies calculus of variation. It is our hypothesis that, if a sufficient number of inhabitants of cities were to recognise urban, peri-urban and rural interfaces simultaneously, yet on an equal basis, one can, at least, avoid some burdens of growth.

In general, in this paper, we will discuss the overall hypothesis that the introduction of a redirected peri-urban land uses system and its continuous expansion can put a wedge between mega-cities and their hinterlands. In our opinion, the testing of that hypothesis should involve proper spacing and land zoning. To this end, the paper provides a theoretical part, looks at evidence from theory and introduces a planning model.

It is the narrower aim of this paper to show how to better study actual contributions of improved strategies in peri-urban land use. The strategies shall sustain regional development of mega-cities and offer scope for public interventions. We will work within different scales of populations addressed as land occupation (below). We will discuss six major research subjects in regards to ecological issues, expressed as entropy (below). In particular, it is our research objective to develop a more integrative thinking of "peri-urban" and make it a self-standing and innovative subject with a focus on planning. A modelling is provided to solve land use planning by referring to links between such sub-subjects as (1) land segregation, (2) food, (3) recycling waste, (4) income and (5) providing space for (6) better living. Yet, it needs to go into the "peri-urban" as a concept. Then still simplified, modelling results in a control problem.

Finally, subjects will be jointly discussed in a comprehensive package for future work (plans), and we will offer suggestions, particularly from a theoretical point of view, on developing a new concept of the "peri-urban". Our studies shall also contribute to a formal approach to policy modelling.

## 2 State of knowledge and recent developments

### 2.1 Examples and site-specific knowledge of peri-urban situations comparing past and present

Over the last decades rural-urban interfaces have been studied intensively. For instance, in many countries, the geographical and socio-economic conditions of peri-urban regions and

their development have deteriorated (see for example Malano et al. [4]). However, studies have mainly been conducted in terms of descriptive and case study analyses. Said studies have come up with issues such as viability of regions, functions, etc.; a general view is that little has been done in order to solve problems for sustainability in a nexus of spatial outlay and sustainability [5]. Originally and with respect to envisaging the general objective of improving sustainability, mainly by stopping growth, the focus of studies was mostly on urban dynamics and identification of gradients in an ongoing process of growth as well as unhealthy spatial change. There are many studies which show driving forces and interfaces [6]. Recently, a focus has been on the urban-rural fringes in terms of food systems and land use [7], and one can find complex suggestions for spatial planning [8].

In particular, a major focus of studies has been on regional aspects in general, such as demographic trends, economic activities, special geographic features, and spatial layouts for peri-urban areas, etc. [9]. Many case studies have also documented local aspects such as rice field reduction, crop and land loss, migration for jobs, land tenure insecurity, etc. Also studies detected community organisation deficits [10] (for more modern cases see [11]). Then, in the south where the "Green Revolution" in rural areas was promoted, there is a lack of specific addressing of peri-urban areas and a chaotic development may also prevail [12]. This actually concerns planning and action needed to establish markets for specific functions of peri-urban areas such as nutrient recycling, employment and food provision. A major problem is to raise demand for wage labour in farming and the recycling of organics, which could provide a livelihood for local households [13]. Yet, some studies foresee chances in recycling of nutrients and job creation [14], but little has been achieved so far.

During the late nineties of the last century, some further major findings already showed that land around cities became scarce and expensive. One example was Indonesia. Especially former rice fields were sacrificed to urban expansion [15]. Mechanization to increase labor productivity and farm size to cope with higher off-farm incomes is a possible solution to this problem. But, with rice land keeping on shrinking due to urbanization, it will be difficult to be independent from food imports [16] which again are expensive.

Moreover, the issue of the decline of peri-urban farming is not a new phenomenon. In the past it was found that income increase(s) is (are) considerable, in particular as remittances from migrant labour [17], and that farming has become less profitable. With respect to regional planning, previous studies [18] have provided an agenda for the planning of service jobs that put an emphasis on public provision such as health facilities, credit access, etc. and discuss the pros and cons, for instance, of credits for sustainable agricultural intensification (also [19]). For instance, as a complement, it was found by [20] that agriculture in peri-urban areas (for example in Indonesia) is on the decline in terms of productivity because of negligence. However, only a very limited number of land zoning and priority setting policy discussions appeared. Instead, White and Wiradi [21] showed that peri-urban land use will increasingly rely on opening opportunities due to regional planning as related to sound spatial economic analysis, i.e. due to food processing as related to raw products from improved peri-urban agriculture, and perhaps due to waste management as related to nutrient recycling by farmers.

In particular, the scope of organic waste management was already mentioned early on by Yayasan [22]. Organic waste disposal and control in peri-urban areas is a major problem. Unfortunately, seemingly comprehensive studies on rural-urban linkages, like the one mentioned before (Rotge et al. [10]), still report few successes with regard to waste management; perhaps because of a lack of local demand by farming. Let us take the example of Yogjakarta, again. To the knowledge of the author there were already investigations in the nineties of the last century. Thirty years ago, in a rough study on Yogjakarta's waste problems, it was

estimated that in UAY (Urban Agglomeration of Yogya) the average waste generation rate was about 3.25 m$^3$/day /person/day, multiplied with the total population of about 930,000, waste production was at least 3,020 m$^3$/day, or 1.1 million m$^3$/year [23]. Today, most waste still goes to landfills, and efforts for peri-urban recycling, composting, etc. are low [24].

Parallel to that, for example in Bangkok, Thailand (as illustration also a quarter century ago), research was conducted on the emerging waste management problems of the town itself and on the disposal of organic waste in peri-urban areas; disposal as a way out of pollution with organic nutrients. For instance, feasibility studies on the potential of agricultural application of sewage and night soils [25] were conducted as early as the nineties. Since urban problems in nutrient balancing for Bangkok [26] were expected to be pivotal to the future, first attempts to select appropriate sites and suitable soils could be found [27]. Many problems have follow-ups [28]. One problem is who demands organic fertilizers. As reported, there have been no changes [29]. Nowadays, even intensive agriculture is a major problem [30]. It hampers progress in peri-urbans, for instance in Bangkok, and creates externalities in unprocessed organics, etc. [31]. To summarize: issues prevail in terms of ecology, planning and policy and in regards to the linkage of peri-urban agriculture and cities.

## 2.2 Ecological economics

The site-specific state of knowledge (above) has to be confronted and supplemented with methodological research on the sustainability of the economic, social, and ecological systems that lay the ground for city development [32]. Moreover, some peri-urban research has been now based on ideas of ecological economics [33]. Authors in economics have shown that ecological principles can enter into the analysis. Moreover, as Daly [34] has shown, making economic systems more self-regulated and self-reliant by recognising macro-effects from micro-level decisions is urgently needed to establish circular flows and recycling-based economies. One question, in particular for growing cities, is to re-consider the throughput of materials. It matters for scaling activities and for designing the spatial outlay of city systems including functions (below). However, ecological-economic approaches and models require a common grounding in the value and evaluation of prevailing systems or paradigms that support integrated system evaluation. Through a spatial approach, the suggested initiatives will allow us to gain insight into ecosystem structure (below). We need joint bio-physical and economic views on sustainability [35, 36]. In particular, since integrated values from natural, semi-natural and culturally transformed areas are urgently needed for decisions making, we may pursue the idea of integrating economic evaluation (monetary) and ecological calculation of system functions (for instance measured and suggested in entropy terms: [37]).

With respect to the question of integrating peri-urban land use in city outlets and fostering sustainable regional development by planning and land zoning, only limited attempts have been made so far to achieve such system analyses because the complexity is high [38]. Though there is literature that shows attempts, implementation has shown limited results [39, 40]. However, attempts to solve the above-mentioned complexity by applying new methods of handling spatial problems date back decades and are puzzling due to confusion. Again, as documented years ago by DeTomb [41], specific spatial outlining bears more importance.

As a consequence, multidisciplinary research on regional planning for peri-urban areas can benefit from recent discussions and experience in urban economics [42]. Topics that are appropriate for discussion are externalities, government interventions, and indicative planning, including land zoning. An analysis by Anas et al. [43] suggests that externalities justify government intervention in land-use decisions. Hence, we need a lower intensity in order to get less pollution. Furthermore, landowners' property taxes may cover the costs of public

goods provision in polluted areas [44]. The most common methods to achieve these ends are land-use zoning, taxing, promotion of regional economies of scale, etc. These approaches have been used to separate pollution from non-polluting activities and mitigate urban problems at the same time [45].

Specifically, in much of the respective literature, land zoning is determined exogenously, not linked sufficiently to regional taxing, land prices or pollution. In contrast, some literature emphasises endogenous zoning. Tomasi and Weise [46] demonstrate that spatially differentiated Pigouvian taxes on emissions are generally not sufficient to ensure a competitive equilibrium being Pareto efficient. Taxes need to be complemented by land-use restrictions. Also Hochman and Rausser [47] confirmed multiple zoning, where numbers and borders of zones were determined endogenously. Modern analyses should combine spatial and dynamic aspects of land-use controlling, as it is important to address problems within a dynamic and spatial setting [48]. It needs a reduction of complexity which we will address in a simplified model below, applying entropy as (re)-novation.

## 3 Need for innovative features and conditions for ecological economic development

Firstly, when assessing socio-economic dynamics, potentials for reduction in pollution, financial flows and new patterns of exchange, one faces the problem of infant diligence. In order to establish contrasts to the current development of heavy energy dependency and entropy, innovations for land use fitting into systems have to be established. The foremost contribution of studies should be to develop viable strategies for improving the life support functions of megacities by integrating peri-urban areas into business flows based on spacing. In particular, strategies must be developed that

- sustain necessary economic transactions for a continuation of business,

- sustain the basis of natural resources, despite an ever increasing potential demand for natural resources and waste disposal by large and growing cities, and

- maintain social organisations, develop mechanisms that contribute to conflict mediation, and proliferate social institutions that guarantee the survival of the poorest in large cities.

Secondly and equally important, the disciplinary focus must shift to an integrated approach of spatial appearance of well defined peri-urban land use system and recycling, that includes points of transaction for services. For example, waste has to be collected location specifically in space and compost has to be distributed and applied on available land. This shall guarantee minimal ecological externality effects, given economic activities. To this end, one can use

- land use planning at specific locations (reservation of land for agriculture, recreation, etc.),

- land zoning over a regional or spatial unit (look at a system of concentric activities),

- get shadow prices which reflect the need for subsidies and development activities (government incentives to construct and finance public facilities).

Thirdly, as interdisciplinary research projects, studies should be devoted to

- pilot studies on innovative concepts, for instance, in waste management,

- experimental design in study areas on more appropriate technologies, and

- checking technology innovations in small-scale industry and agriculture.

Fourth, one has to look into bounded rationality of material balances and recycling, find a unified approach based on ecological economics and show how it can be applied. A scientific framework on ecological rationality has to be developed, minimising overall degrees of external dumping.

Fifth, one may want to know how peri-urban land use will contribute to food security, local, regional food markets and value chains. For that we suggest simulations built on land use planning (below). A spectrum of detailed regional impacts of peri-urban agriculture, distribution and macro information on food chains is envisaged in order to design targeted policies for a sustainable layout.

As the situation of sustainability has changed drastically (it has declined), new less energy intensive and but more labour productive technologies are to be introduced. For that we also need simulations. Diagram 2 (below) outlines the basic hypotheses on current intensification and we suggest modifications of spatial economies in mega-cities in Diagram 3. Our hypothesis is that the recent economy is strongly energy dependent, and this has considerably increased the energy use gradient between urban and rural areas. Although agriculture in rural areas has also intensified in terms of energy use and now produces more waste, such as nitrogen leaching, etc., it is lower in entropy than that of a city and can convert organic wastes into organic fertilizer. In terms of recycling urban waste, peri-urban agriculture still has the potential of contributing to efficiency in the use of energy and lowering of entropy. However, some levels of welfare should be sustained and we already see, for instance, large-scale farm operations.

## 4 Theoretical deliberations on sustainable urban development

As has been mentioned, a basic idea for a unified concept on ecological impacts of peri-urban land use stems from reducing energy use and minimizing negative entropy impacts. Entropy can be considered as waste from human activities, and ecological systems are sound if entropy (opposite of chaos) is low. Entropy has been presented as an indicator for ecologically unhealthy developments for human-dominated eco-systems [49]. It goes beyond energy use, but includes energy overuse (mainly fossil energy) as a major threat to sustainability. In regards to mega-cities, developments have been criticized and we further reference studies on urban health studies, putting the energy aspect of entropy into focus [50]. Also, spatial distributions, origins and flows of energy and finally entropy of the system "city" have been methodologically discussed and visualized [51]. For a flow analysis, typical for cities, it is important to take a look at regional association in order to get mitigation and interventions operational. We will make some theoretical and systematic deliberations and put them into a spatial frame. The idea is to attain abstraction, conceptualisation and indication of cities by a radius-driven insight as starting points built on an indication for planning.

### 4.1 The ecological dimension

We start by comparing traditional land use and recent observations on city sprawl within a range from city centre to periphery. Diagram 1 sketches the idea of a traditional multi-location oriented organisation of a region that has had a small urban centre, narrow peri-urban strips, and a sprawling agricultural periphery. The gradient can be depicted by a radius, if the city follows more or less a cycle structure [6]. Land use is presumed to be organized around cycles which can also be put into a continuous perspective. The radius, in Diagram 1 and displayed as x-axis, represents the spatial dimension (size and stretch) of the city. On the y-axis, we display the negative impact (entropy as subject to externalities). As seen above, of a joint regional system, it is low compared to Diagram 2. Further note, the bowed arrows show local recycling.

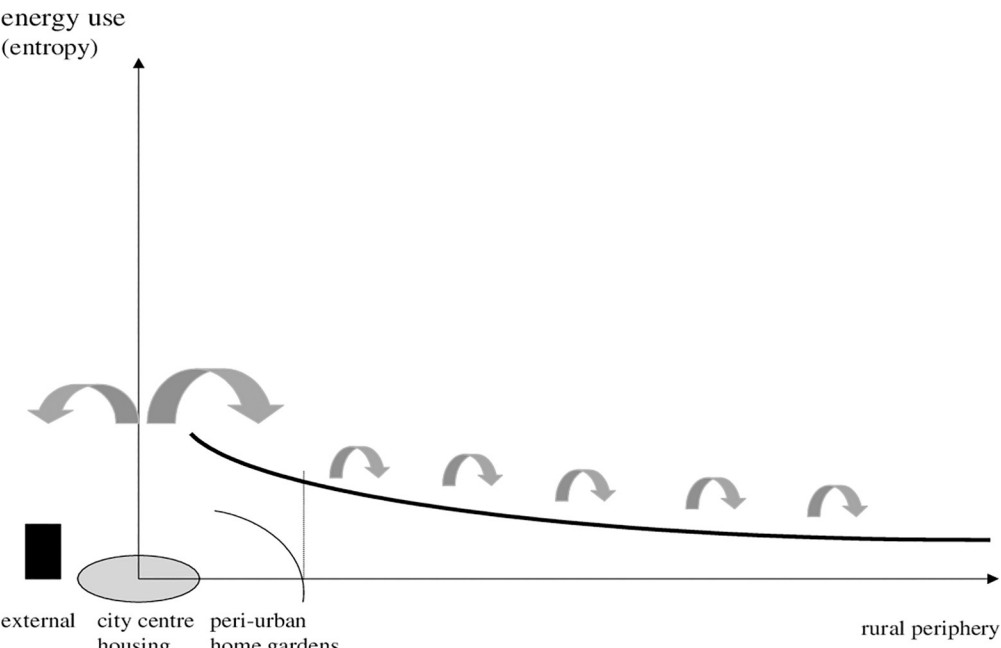

**Diagram 1. Traditional gradient of ecological impacts.** Where: arrows show recycling, area of city size is shaded, the city dimension is given as continuous from spread as in a circle economy [58]. Source: own design.

In traditional or pre-modern systems we observed a gradual increase of energy use from urban to rural areas, but levels are moderate and it seems that nature has tolerated human activities because of scale. Usually we focus on energy as opposed to entropy. A consequence of rapid urbanisation is a dependency on external sources (oil, fossil energy or fertilizer) and less

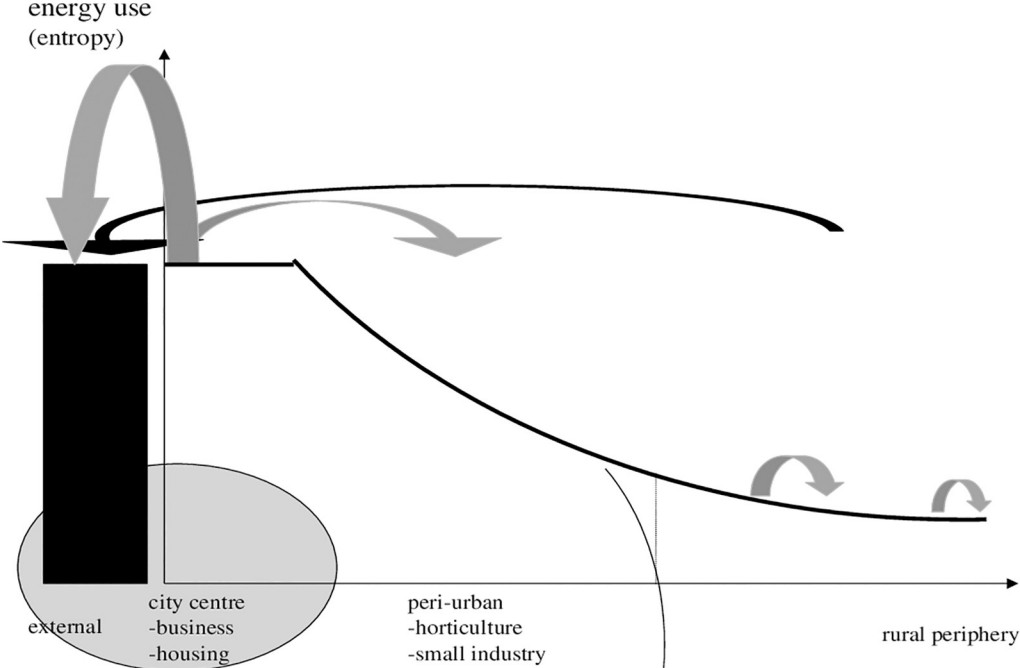

**Diagram 2. Modern gradient of ecological impacts.** Where: column: external sink. Source: own design.

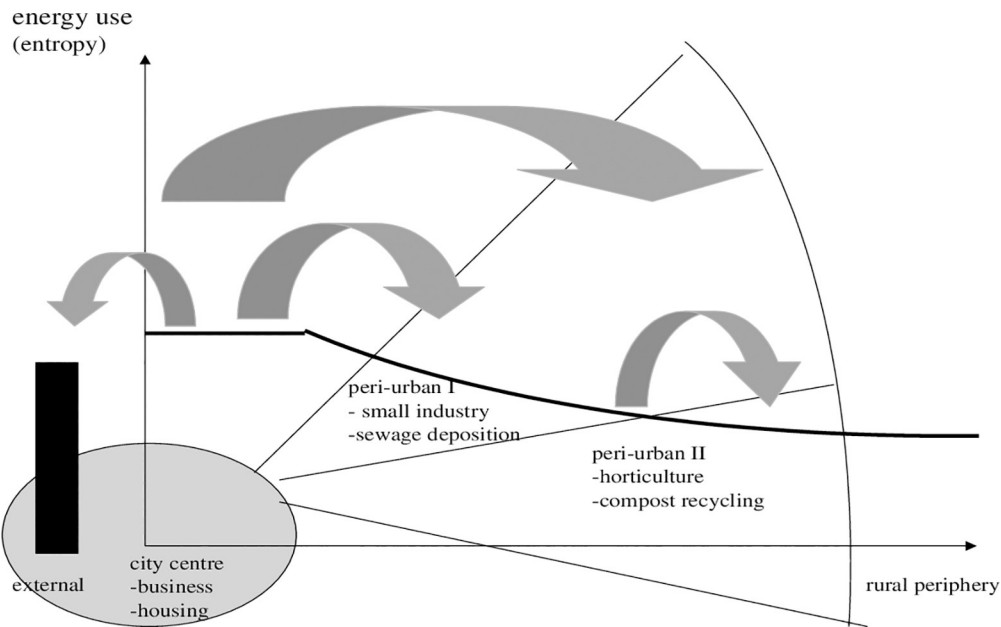

**Diagram 3. Gradient of ecological impacts of peri-urban land use.** Where: circle: peripheral effect on rural fringe. Source: own design.

recycling of energy. That is, from a thermodynamic perspective, negative (entropy) and includes farming; farming can, however, cause lower entropy by recycling.

In Diagram 2 the city is bigger (shaded cycle, representing urban land, increased), huge waste disposals occur as externality and count for a negative impact on entropy bent by city development.

In this case, what is the scope for land use planning and change? We see organic deposition, food and horticulture production as options. In Diagram 3 we suggest that for a functioning peri-urban land use system, which is based on recycling peri-urban farming. Farms or regional districts with peri-urban feature should expand (eventually labour intensive and organic, though this is another subject). Thirdly, referring to the introduction of multiple functions that could be provided by peri-urban areas, a special focus should be on organic waste management, sewage treatment, and sludge processing to make organic urban waste applicable to specific types of peri-urban agriculture, etc.

Change in land use can reduce entropy. Basically, we consider three aspects of material balances: recycling, alleviation of entropy derived from external sources and less transport. This will include peri-urban cropping and livestock, food processing, merchandising for local markets and green corridors. In this context, a link between food consumption by peri-urban dwellers and food production becomes relevant.

On lowering entropy: First, local food production by peri-urban agriculture should have several features and characteristics which are subject to locally deliberated policies: it should be low in entropy generation and should offer healthy food. Yet, local food production is complicated with respect to entropy; it needs land and less fertilization. It is not always better than imports and scales matter? A system approach is needed in regard to food production and consumption to qualify (minimize) energy use and offer nutrient recycling options as dependent on spatial distribution of process and consumption. One modern question is, for instance, how a proliferation of supermarkets and outside sourcing of food as well as imported food in peri-urban areas contribute negatively to ecological goals. So, we enter into economics and

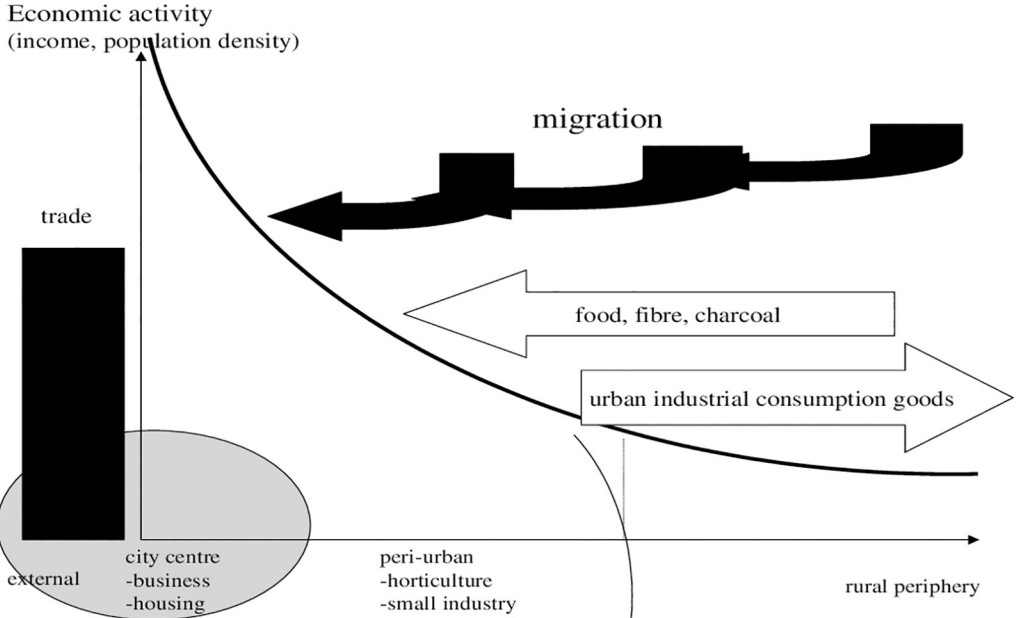

**Diagram 4. Modern gradient of economic activities.** Source: own design.

must reveal queries on "right" farming. For evidence see references and examples for recent discussions on the spread and dynamics below.

## 4.2 Economic dimension

This paper takes an additional spatial perspective on economic and regional development in the rural-urban interface. Accordingly, to the three scenarios above, we can simultaneously structure the economic dimension in a range from traditional mode to an established urban-rural interface (given in Diagram 4). The current mode is shown in Diagram 5 and the

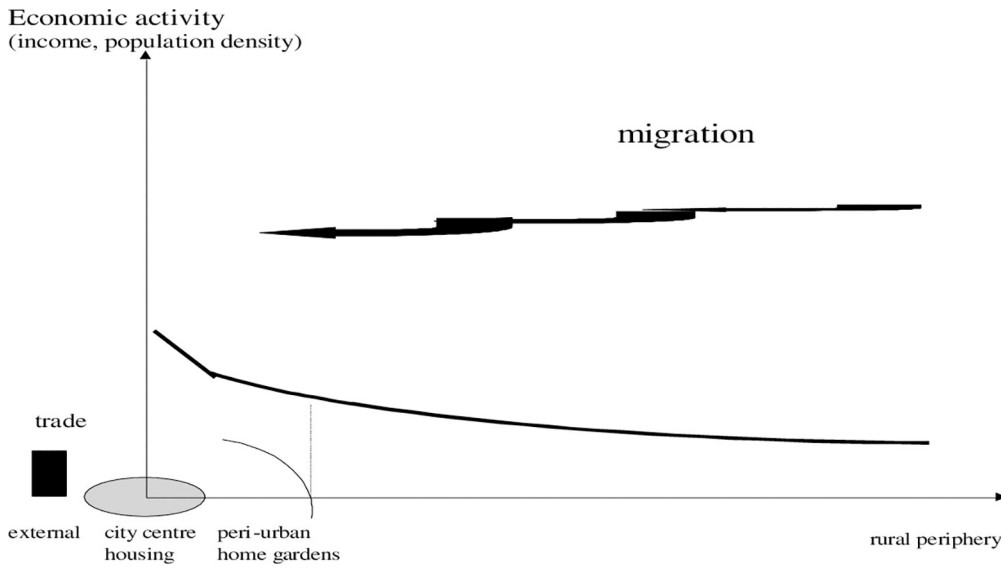

**Diagram 5. Traditional gradient of economic activities.** Where: economic activity: index. Source: own design.

recycling variant in 6. Migration was noticeable in the past, which we call traditional. Incomes in the rural periphery were low, though a gradient existed and some migration was pertinent. Incomes and in particular food production were so low that some people spoke of rural slums. Vice versa limiting city growth was seen as costs and policy instruments were supposed to minimize costs. "Peri-urban land use" was an option. Large parts of income (as non-monetary or food) was generated from agriculture and spent locally. Reckoning these aspects in a gradient of economic activity which included some trade, a traditional Thünen-model with high transport costs applied and it was at the edge of change.

A major question is: Which economic role played by peri-urban areas concerns income and want are costs for its implementation? In this context population can be regarded from an economic perspective. Income is provided by further urban industrialisation and population increase can be economically accommodated; that was the idea of city growth and farming lost. But income can also be generated by peri-urban farming. The question is: Will the market do it or do we need planning? In the past, developers have overlooked specifics of peri-urban areas because food was cheap. One can formulate the hypothesis that economic gradients will flatten, then, at least due to "more natural processes" of growth and local food? Perhaps policy will lower migration to centres?

## 4.3 Alternative

Diagram 6 explains how the introduction of a rural-urban interface, which is constituted by publicly designed zoning for peri-urban land use (farming), will change economic activities. Essentially, we pursue the idea that special types of peri-urban agriculture, exposed to waste treatment, recycling and raw material deliveries, should be located as a wedge between urban and rural areas. We emphasise the proliferation of food generation primarily as home gardens, regional exchange of vegetables, etc., but now with an emphasis on jobs and income generation subsidized (see below for modelling).

A modification of the internal spatial organisation of economic activity, accompanied by technologies based on local resources, shall improve competitiveness of products and comparative advantages.

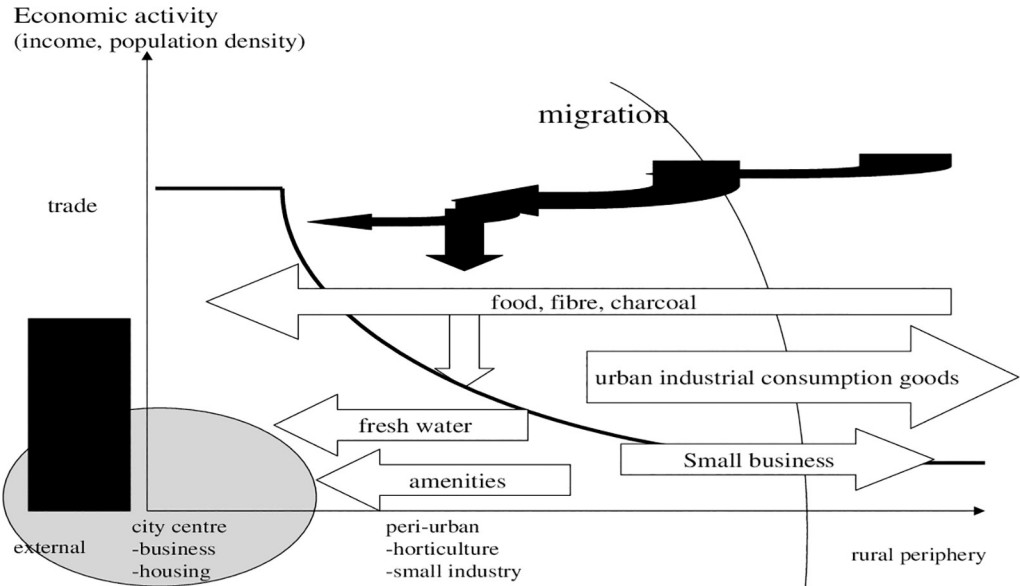

**Diagram 6. Gradient of economic activities within peri-urban land use.** Source: own design.

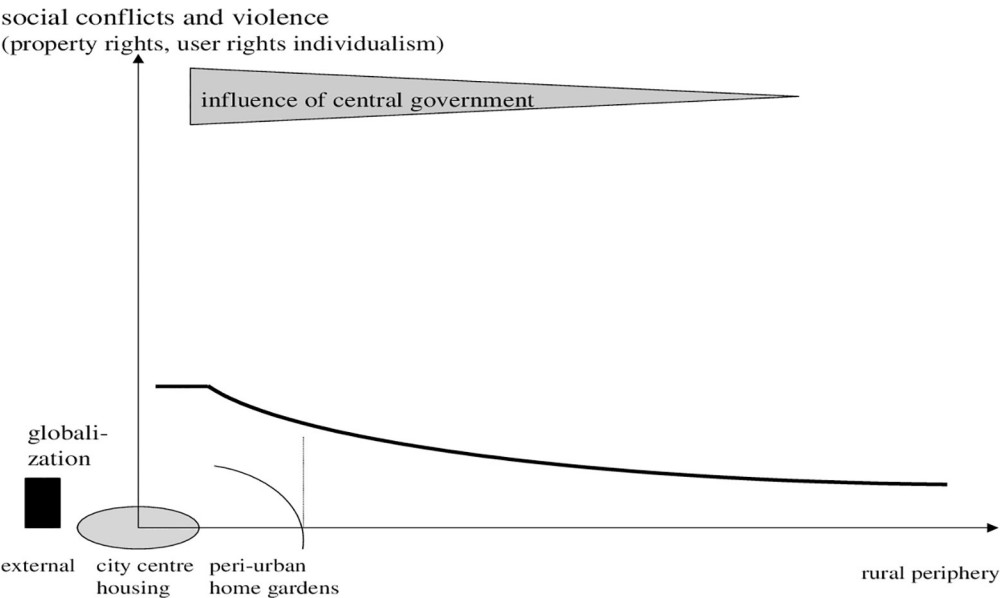

**Diagram 7. Traditional gradients of centre involvement in conflict resolution.** Where: circle: peripheral effect on rural fringe. Source: own design.

## 4.4 Social aspects

Notably, sustainability has three dimensions, of which we have so far discussed the ecological and economic sustainability. But we should also be interested in the minimization of social conflicts that originate from urbanisation. In this regard we need to consider basic needs such as food, health and sanitation facilities as well as we have to accommodate populations (see below as expansion of modelling residence space). Planning for farm land and regional food is figured by improving biases against poor people. Governance of social conflicts is related to land and planning. With respect to the influence of local governance and central power, the peri-urban fringe or interface brings to bear an interesting feature into agency if land claims are grounded in rational decisions. Diagram 7 gives a pre-modern (traditional) set of institutions, conflicts and government influence.

In Diagram 8, a duality of peri-urban areas emerges: rich people live in the city, poor at the periphery. The influence of governance declines. Socially, the question is: Do we have an opportunity to improve the living conditions of target groups which make up the poorest segment of urban population by offering them regional food and occupations in farming? In this regard, globalization is both an indicator and a threat.

Some answers to such questions as, for instance, a reduction of threats of globalization can be observed in Diagram 9. Directing migration and offering public schemes in peri-urban areas are options. By a continuation of migration, but now in peri-urban areas, we may solve land conflicts of growing populations. Apparently, all of these contemplations or measures to solve conflicts are part of a variety of plans.

The ability to solve conflicts depends on the ability of a new, as yet to be developed system to absorb population in job markets and guarantee economic prospects. In our modelling (below) we will specifically address that. Still, the more political question is: Which institutions are associated with peri-urban activities and what is the optimal choice with regard to decentralisation of political participation and fiscal policy? With respect to the overlapping of central and decentral governance, Diagram 9 provides hints on how one may envisage a research

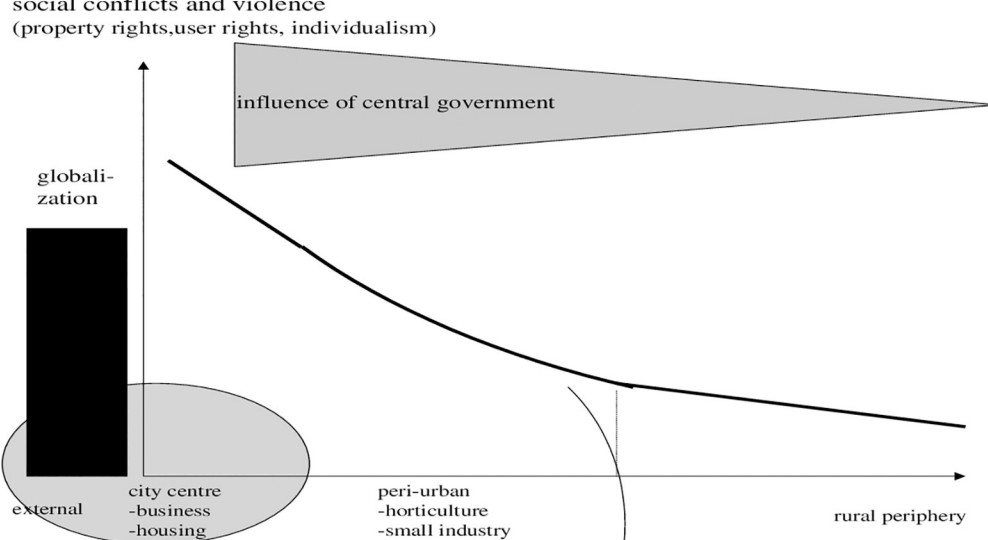

**Diagram 8. Modern gradients of centre involvement in conflict resolution.** Where: wedge: potential policy. Source: own design.

agenda linked to planning. Concerning these issues, in particular regarding the influence of central and national government as well as regarding a decentralisation of decision making, supplements are needed.

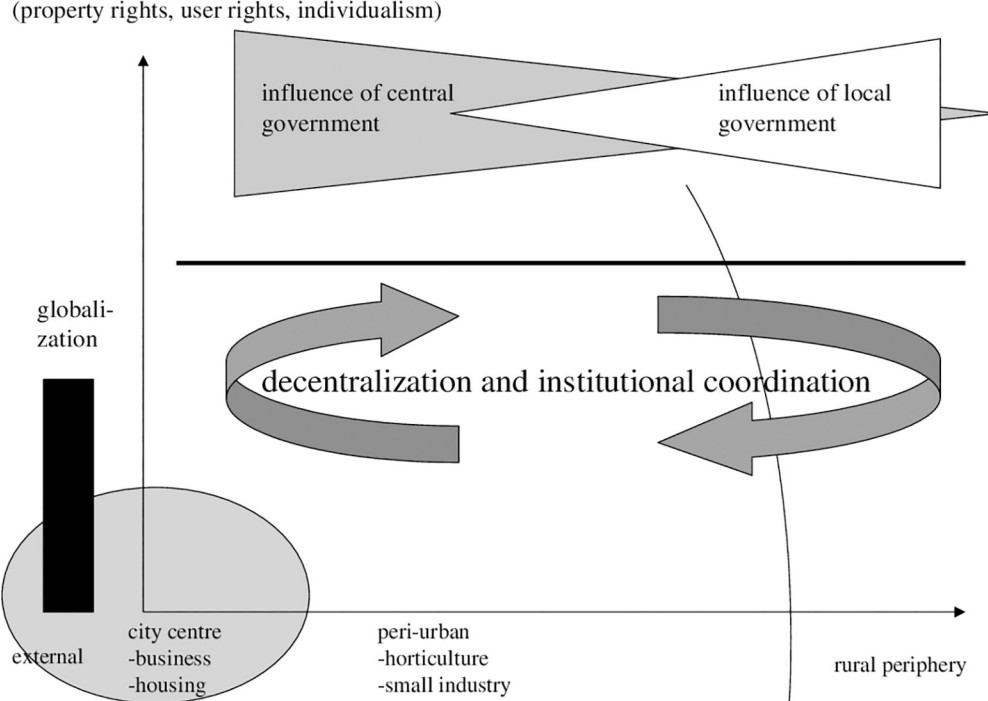

**Diagram 9. Post-Modern gradients of centre involvement in conflict resolution.** Where: wedge: effects and policy related. Source: own design.

## 5 Sustainability, planning and piloting of projects

### 5.1 Likely aims

The aim of regional planning and creation of land entitlement for farming as well as the scoping of specific types of peri-urban agriculture should be to contribute to the alleviation of ecological problems. A new system of land use is envisaged. This system change shall bring about a more sustainable pattern of growth that is less harmful to the environment and promises to reduce the threat for un-sustainability. For indication we suggest a sound ecological measurement of current problems: entropy. In general, in the article, we introduce a peri-urban land use wedge composed of farming between residence and industry (below). Spatially, a continuous expansion as a wedge in mega-cities and their hinterlands, respectively, shall improve the sustainability and livelihood of large cities or agglomerations (entropy). The testing of this hypothesis will involve an elaboration of necessary components for a viable peri-urban food provision to partly feed the population. As related to city-specific needs, proofs of practicability via a simulation of the compatibility of system versus incompatibility of the current strategy of industrialisation are important. Planning counteracts uncontrolled growth. After that, piloting can start, though it needs land to be assigned as a pre-condition.

We recommend to moderate structural change by land use planning, including the development of a peri-urban interface as a special type of food generation based on recycling and local exchange. Labour-intensive farming may be necessary to cope with migration and to find new patterns of lifestyles. The food thus generated is also needed to meet a growing urban demand. Changes to currently resource-intensive lifestyles must be found and proposed on the basis of nutrient recycling. There is scope for organic food, gardening and food diets serving local settings, which is preferable to the adoption of foreign food technologies.

### 5.2 Examples, comparison to recent research and implications for planning

In order to develop ideas on the linkage between regional planning as well as on piloting and for the comparison of projects, the author suggests taking a look at recent investigations on mixed land use systems in agglomerations. They prevail, for example, in China [52, 53]; there, specifically concerning the theoretical background, projects aimed at mitigation of ecological disasters and having a focus on pollution and the decline of farmland are put forward. They also have looked for practical advice and indicators, but may lack constancy. Moreover, we see examples from Africa in which mapping and categorization for peri-urban-farming has been conducted [54, 55]. As the cases are also piloting projects, the above theoretical aspects reappear many times, but are not yet fully explored. Concerning the status of peri-urban land use, one can conclude in general that there is a lack of spontaneous emergence [56] and that planning is needed [57].

It appears as though one needs phases of simultaneous planning and implementation, rather than follow the strategy of doing anything ordered by different government units. For example, we refer to the social science problems of getting local inhabitants involved, of providing food [58] and of getting communities involved [59]. It seems difficult to ensure the participation of locals who should develop a perspective living in a peri-urban area vs. public interest which goes beyond the specifics of certain places [60]. Again we refer to the scope of planning as a tool for simulations which in turn shall help governance [61], for instance, city planning at a spatial scale. Note that in the author's opinion, planning is more important than a simple allotment of land, although it is a first step as will be shown below. What we need are discussions on the backgrounds for local managements, involving theoretical needs as an integrative approach.

## 6 Interim discussion

In the previous chapters, the importance of an integration of farming has been outlined with regard to planning (peri-urban land use) and to having a focus on nutrient recycling and regional food provision. Although we took a theoretical analysis, practical solutions for planning were touched upon; a major question remains, though: How are we to implement the ideas [62] as regards to space? Because of limitations, here, we can only refer to planning initiatives which are mostly concerned with spatial outlays. Then there is a follow-up because spatial priorities give frames for scoping and piloting of business. Still, in business the economic frame of local prices is important. We refer to land prices as being part of planning. Land prices, as such potential or evident, are revealed via shadow prices of land and related to subsidies vs. taxes on land (below).

Furthermore, with regard to technical aspects of planning, the author would like to make the reader aware of developments concerning instruments for public planning at the urban rural fringe [63]. We have to deal with boundary conditions for city spreads in the hinterland and external flows. A fruitful tool is again the spatial control theory (calculus of variation, see below). Here, planning includes flows of materials, stocks and payments, and the spatial analysis for boundary conditions. They supplement flows within the system. It delivers terminal conditions. Note that a continuous city model, yet from centre to periphery, is only part of the picture. Such models exist, though mostly for amenities [64]. Now, as externalities of industry and their disposal in space they are to be discussed [65] with more thresholds. Many thresholds emerge and request regulations at the system level. However, as options for regulations and application of policy instruments, so far, modelling did not include promotion of peri-urban agriculture; new rules have to be made clear by way of simulations. A special topic is local food production.

This means that in terms of mitigation and provision of food, food should have different appreciation in "distance to periphery". Hence, in terms of governance, instruments (tax and subsidy) apply. A major limitation of the systematic flow and regulation analysis is an outline of contingency of instruments in a multidisciplinary mode. This requires us to establish objectives of the social-economic system "peri-urban" (see below), yet in a concrete manner. Planning, as usual, becomes part of an ecological foundation for a city beyond pollution, and food is important in order to nourish the metabolism beyond external sources. We have to contribute to the point of autarchy.

## 7 Method

### 7.1 Spatial city outline and graphical presentation

In this chapter we propose a continuous space modelling as a method for regional planning, especially in order to minimize entropy; it is built partly on emission accounting of sectors in urban agglomerations and partly on density, thus determining the overall picture. We stylize our modelling and see sectors organized in space stretching from centre to periphery. Since, conceptually, cities are circles, the radius is sufficient for an indication of spaces and location. Since the aim is to reduce regional and accumulated negative effects on entropy given a cost minimization of activities (below), integration over a radius is our methodological approach. Space is divided between habitation (residence), land occupied by agriculture (farming) and industry (industrial plants), all as shares in a circle mode of outlay (space picture: Diagram 10). At the beginning, agriculture occupies small amounts of land, which is mostly located at the periphery.

As we work with a continuous modelling approach, the radius displays the dimension of the city under investigation. The radius is given by the x1 and x2 axes; depicting space in a

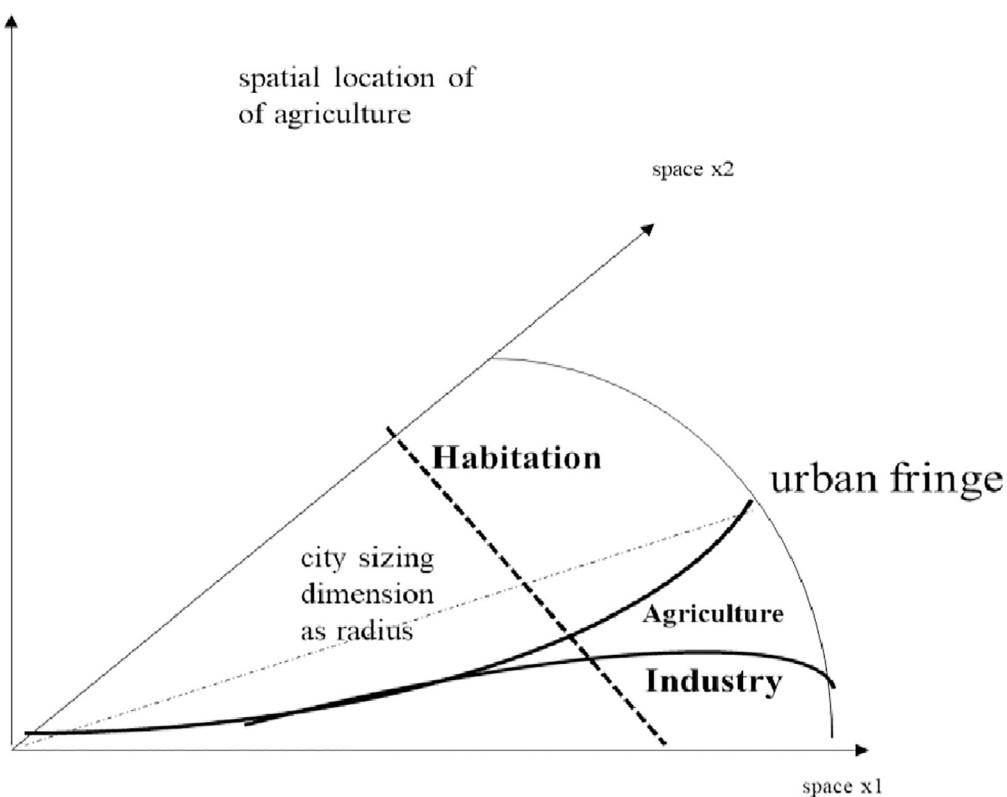

**Diagram 10. Spatial outlay for planning.** Source: own design.

transformed Euclidian way. For a similar approach see Nuppenau [64] in which amenities are displayed also by distance. Now, as we work on reducing negative externalities which means that we are promoting the prospect of peri-urban agriculture, farm land should expand. Farming is displayed as a regional slice between residence and industry (wedge: Diagram 10). Land use volumes are given by functions separating the sectors. (Functions will be introduced soon, see below.) The edges of occupations (residence, farming, industry) are outlined in the x1 and x2 dimension. In order to sketch out the positive effects of agriculture, which are later expressed as four mathematical functions, we refer to Diagram 12. The outline reduce all to two-dimension: radius and effects; preference and effects are treated similarly, because we take x1 and x2 as radius. Areas are measured a long distance by integral and act as system triggers (Diagram 11). Increasing agriculture reduces accumulated entropy. Peri-urban activities matter for the slope of negative effects, and we will analyse what is an optimal choice in space regards to control of political participation and fiscal policy. Note that agriculture is stylized as a wedge, though it may coincide with habitation in special ways.

The processing industry is spatially linked and entropy is also reduced due to green corridors, though differently. Industry is residual and adjacent to farming and residence providing waste. The food connectivity is part of waste (recycled again below). Right now it is only a spatial outline to be supplemented. (The detailed way of coexistence is a question of local outlay, above.) In Diagram 12 the effects of Diagrams 10 and 11 are synthesized. For the sake of showing effects on entropy, the space of peri-urban agriculture shall be increased (soon optimized, which is the task of the next chapter). The diagram shall indicate that entropy can be geared by land use planning.

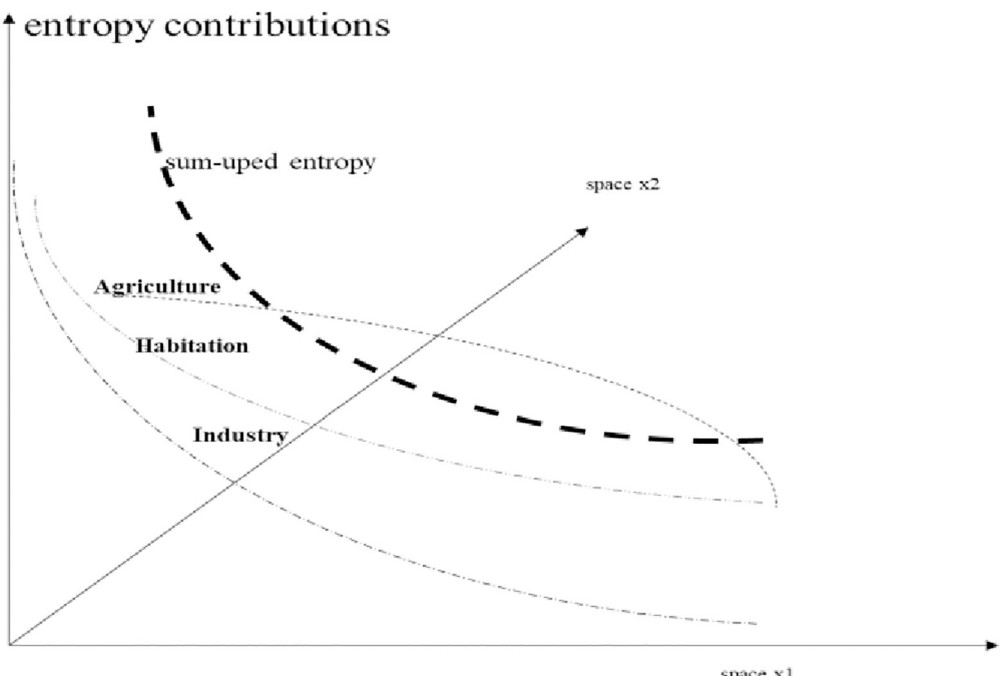

**Diagram 11. Contributions to entropy by distance and density.** Source: own design.

The diagram shall indicate that entropy can be geared by land use planning and policy instruments such as subsidization of farming and taxing of industry as well as habitation. The purpose of mathematical modelling (research of optimal spacing) is to find the slope of functions. (Below we display the mathematical outline.) For empirical research's sake, functions can be retrieved from a city serving as an example.

Note that the instrumental effects have to be included in shifts of functions as intended by planning. Planning shall be based on economics in terms of returns from farming, industry and residence.

Also keep in mind that we need the current city outlay as a reference and that it is quite evident that individual cities can be different in spatial outlay and it needs abstraction. However, any city can provide needed data to regress the functions (edges) in space. We must reference to given situations. To this end, spatial regression is appropriate and a well-introduced tool, although we have to find observation and points in order to achieve a coverage of space and entropy distribution. Subsequently, we have to mix spatial outlay with empirical evidence on the profitability of land use categories (if sectors are given [64]). As reference, a pure private and land market oriented system may prevail. Moreover, we postulate that citizens to live close to the centre (second y-axis).

Admittedly, we suggest a quite simplified set-up (Diagrams 10–12); but it is feasible to work with such framing [65]. Nevertheless, to reduce complexity, but also to gain further flexibility real cities are squeezed. Then the idea is to work on land categories (assignments) and elaborate on taxing as well as subsidization to get a wedge as opportunity for farming. I.e. if land prices vary (competition: [64]) planning [65] is due to incentive design. It will indirectly gear city development and entropy.

The issue and tool for planning and incentive optimization is land sharing, which shall enlarge land for farming towards appropriate distance; perhaps periphery. Given a formal approach, boundaries (edges) become functionally detected (below). Also, in Diagram 11 we

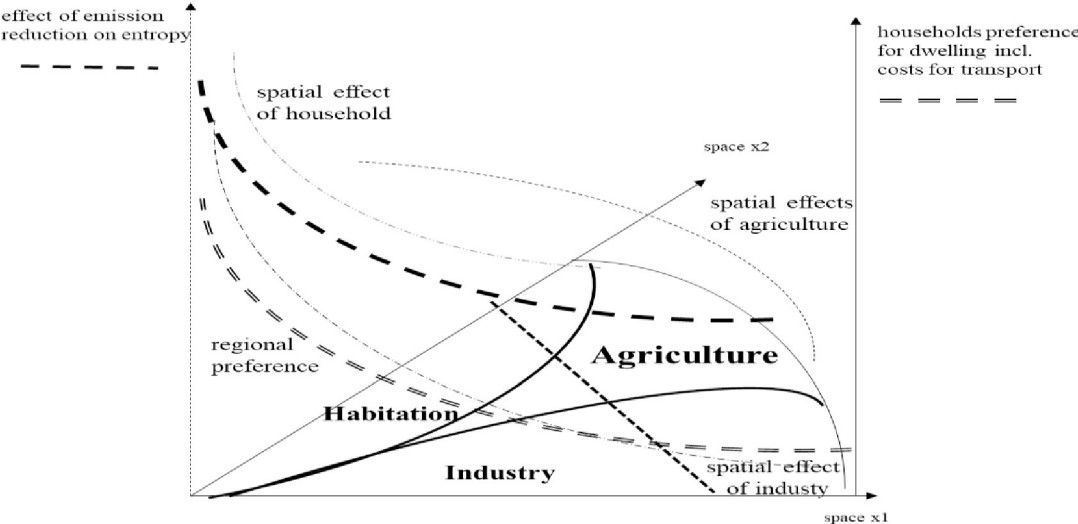

**Diagram 12. Spatial outline, functional effects on entropy and households' preference.** Source: own design.

included a preference of citizens to live close to the city centre, represented by a second y-axis and being part of the later model on continuous welfare summing up. It appreciates people's economic wish (willingness to pay) for high standards of living closer to the centre; for instance, because of lesser transport costs, more pleasure as measured by opportunities, etc. We must conclude that such preferences, yet in real mega-cities, are realistic. They prevail regards to driving forces in social processes of urbanization. Still, it is an empirical question to work out the slope of preference functions. In order to reduce complexity, the preference is given and generalized; over stretch from centre to the periphery. Other slopes (issued in Diagram 11) are changeable through land planning as instruments.

## 7.2 Planning as control theory approach—The system performance

To a certain extent, we follow the well-introduced formal concept of Wilson ([66] from as early as the eighties) for modelling; but try to operationalise and modify it for differential equations and control [67] in space. This is possible if we use the radius as dimension. As outlined, it was suggested that one way of analysing and improving entropy (reducing emission effects) is spatial planning. This shall deliver policy instruments promoting peri-urban agriculture as a continuous function approach from centre to the periphery. Now, given the above frame of planning which is along the dimension "distance from centre" (radius; i.e. technically we suggest to use functions as below for spatial outline and entropy), spatial planning can become a control problem or calculus of variation [67]. For control's sake, the planning dimension is distance and not, as usually introduced, time. Diagrams 11–12 gave distance as radius for city occupation. Then, as we address a city by distance from the centre to the fringes, the overall spatial outline of instruments becomes evident.

In between notice that control theory is a tool for planning. Our work focusses on indirect power given through control theory for land use in farming (enlarging of peri-urban agriculture). The tool is subsidisation of land and taxing industry and residence. We specifically work on the recovery of positive effects of supporting peri-urban farming. Entropy is recovered through land use and recycling of organic matter. For instance and basically, now, the above circle economy is put into a mathematical model exploring improved spacing. Spacing is

outlined as shares for (i) habitation, (ii) industry and (iii) agriculture. From an environmental economic point of view and for urban agglomeration this means that industry and habitation create welfare, externalities and losses, which are to be minimized. Agriculture provides food and recycling as a service. It is publically optimized by spacing.

Diagrams 11–12 has already given us the spatial outlay, which is now translated into a mathematical, i.e. functional, description. As said, for analysis we used a segment of a circle which is shared and whose radius is put as dimension of area size. Actually, we construct shares by distance (length of radius) and angle (i.e. 0.5 times the x1-x2 dimension). This offers mathematical an approximation such as in Eq (1). Note that for total land sharing which is based on the sum of exponential functions (vice versa the angle is given and one occupation is residual) we take integrals and relate shares to the parameters discovering the outline.

Initial land sharing:

$$a_f(\iota) = \Theta_{1,0f} \cdot \exp \Theta_{1f} \cdot \iota = \Theta_{0t} \cdot \exp \Theta_{1t} \cdot \iota - \Theta_{0i} \cdot \exp \Theta_{1i} \cdot \iota - \Theta_{0h} \cdot \exp \Theta_{1h} \cdot \iota$$

where

a area, f is suffix agriculture, suffix h household, suffix i industry

$\iota$ distance

(1)

Having mathematically outlined the area occupation, which can be statistically regressed or based on land use modelling (hereby in discrete step), programming (incl. prices) serves to get an Eq (1). It is achieved with testing instruments in simulations, whereas s(ι) is a planned subsidy for farm and t(ι) an emission tax.

$$a_f(\iota) = \Theta_{0t} \cdot \exp \Theta_{1t} \cdot \iota - \Theta_{0i} \cdot \exp \Theta_{1i} \cdot \iota - \Theta_{1,0f} \cdot \exp \Theta_{1,2h} \cdot s(\iota) - \Theta_{1,0i} \cdot \exp \Theta_{1i} \cdot t(\iota)$$

where

s subsidy for agriculture (indices on labour, land, ev. for special food price such as staples)

t for industry (households) on emission (reduction of regional entropy)

(1′)

In the amended case for land sharing, instruments s(ι) and t(ι), by which the complete system is optimized, are retrieved; they are local-specific, i.e. differing in spaces. Mathematically it offers

$$a_{f,n}(\iota) = \Theta_{1f} \cdot \{\exp[\Theta_{1t} - \Theta_{1h} - \Theta_{1i}]\iota\} \cdot \{\exp [\Theta - \Theta_{2h} s(\iota) - \Theta_{1i} t(\iota)]\}$$

(2)

As approximation of agricultural land use within a segment of a city, which can be controlled by tax and subsidy (for spacing peri-urban agriculture), Eq (2) is part of control. Vice versa the coefficient for agriculture is residual if the functions for habitation and industry are established.

### 7.3 Entropy mitigation

Until now, we have discussed area controlling. The next step is to embed area in entropy controlling. The issue of entropy and emission reduction as well as of organic nutrient recycling with respect to land use has to be clarified further, here, in order to apply the method of control theory. We reckon entropy as a composed regional result of land use and technologies (including resident density and industrial metabolism [48]). As we want to mitigate the entropy of the system, we have to synthesize the overall entropy. From the depiction of framing (distance to centre and local entropy: Diagram 11) it is apparent that entropy is lower in greater distance than towards the centre, i.e. we refer to the description of urban metabolism. —Reckoned from the notification of this article, entropy is impacted by emission and

industrial metabolism, incl. contributions of residence, dependent on distance. Entropy is a state variable which is measured negatively; agricultural land use is positive as less food is imported and organic matter is recycled. I.e. naturally higher entropy prevails in more rural areas and decreases with an exponential trend due to urban activities. This means that along definitions of exergy use (further defined as negative entropy, see above, and externality) entropy and the system become newly composed. Thus a (mega-) city stretches from the lower and from the higher end plus from all levels in between; it is incessantly framed.

Subsequently, as the subject for control entropy is minimizing the difference (increase), again between the natural and the prevailing system, it matters how land use changes. The composition for entropy and quantity is variable. Basically we address a negative externality [65] by optimization. As we want to orient planning towards a prescribed target of the total system (negative externality), a major question is the following: What are the synergies and how can they be numerically depicted over space? In regards to this, what shall be flexible and what is given? We take a differential Eq approach (3) to depict distance $\tau$ to the centre which enables stocks and flows. There is a distinction between entropy as a stock and local injections as flow variables; both are displayed at any distance from the centre to the periphery. The flow variable is local "gain" through farming; the next distance is stock change (3). For food intake we claim that farming has an impact on city entropy by more local food and recycling of organics; habitation is still important.

$$\dot{E}_h(\tau) = \varepsilon_{h,1} E_{ih}(\tau) + \varepsilon_{h,2} a_h/a_t + \Theta_{1,0h} \cdot \exp \Theta_{1h} \cdot \iota$$

where

$E$ entropy, suffix $h$ household

$a$ area, suffix $f$ farming $t$ total, $i$ industry

$$(3)$$

Having explored the frame for planning of entropy and having defined entropy as a location specific mean, incl. accumulation, resp. mitigated by units of agriculture, the sharing of total $E_s$ is given by

$$E_s = [a_i/a_t \; E_i + a_h/a_t \; E_h] - a_a/a_t \; E_a.$$

$$(4)$$

Area sharing in Eq (4) shows entropy as being subject to specific entropy of occupations and shares in land. This indicates possible location-specific changes by change in area and unit rates.

It can be assumed that in residence and industry occupation specific entropy levels are constant due to practice. In contrast, entropy declines with regard to agriculture more if soil nutrients are recycled and agricultural area in peri-urban increases. We presume a function (4') which fits differential equation outlays. The intention is to achieve (5): a display in terms of space and enactment.

$$E_a(\tau) = \Theta_{0,f,E} \cdot \exp\{-\Theta_{1,f,E}[\xi_f a_f/f]\} \tau$$

$E_i = $ constant, $E_h = $ constant

where

$f$ food needs of inhabitants

$$(4')$$

Food needs, respectively, are determined by the population at sites $\iota$ which is area coded.

Hence

$$a_f/f = a_f/\xi_h a_h \tag{4''}$$

The double dividend is coded in land use. However, the coefficient of food choice is endogenous.

For system performance, which shall be total entropy $E_s$, yet along gradients, we finally claim that system dynamics (regionality) is described by a combination in terms land use shares for i and h:

$$\dot{E}_s(\tau) = \varepsilon_{s,1} E_{ih}(\tau) + [\varepsilon_{s,2} a_f/a_t + \varepsilon_{s,3} a_i/a_t + \varepsilon_{s,3} a_h/a_t] + \Theta_{0s}\cdot\exp\Theta_{1s}\cdot\iota$$

where additionally $\tag{5}$

s system exposition; $\iota$ combination of space and distance

By way of explanation: since there shall be a double dividend (positive impact) from farming for entropy, the coefficient of land use in agriculture shows synergetic effects. The impact becomes endogenous once the operational size of industry/habitation is lessened. This is due to taxing of entropy and subsidization (below).

Let us briefly contemplate: entropy taxing at distance $\tau$ has two implications: first, land use changes because relative land prices are impacted and land rents adjust. This happens if land rent is taxed for both industry and households. The tax is used for subsidization. Implicitly it subsidises farming. This means that we can expect a response in entropy by structurally changing the composition of land use and fiscal policy. Secondly, entropy (since it is a criterion for tax) will be reckoned as priced input. Finally, entropy depends on subsidization s and taxing t. The issue here is: What is the change by which the two policy instruments can lower entropy in industry and households respectively, i.e. lower the total externality? We claim, a reduced form for functions delivering (6):

$$\dot{E}_s(\tau) = \varepsilon_{s,1} E_s(\tau) - \varepsilon_{s,2}s(\tau) + \Theta_{0s}\cdot\exp\Theta_{1s}\cdot\iota \tag{6}$$

Differential Eq (6), which is based on distance $\iota$, outlines a control system [64]. (Perhaps it has to be explained in depth by interactive equations derived from objectives used in the analysis).

Note that for empirical estimation and a deeper and detailed analysis, a commencing problem is the display of the linkage between change in entropy and nutrient flows to agriculture. Waste recycling and food from agriculture are retrievable by simulation, of which we will get results. Also, having achieved inclusion of recycling agriculture in peri-urban land use, it can be presumed that imports are replaced by local food (an aim beside entropy reduction), as recycling of soil nutrients close cycles. Then, assumptions about types of agriculture have to be made. We postulate recycling of soil nutrients, also such as from imported food, and enriching soils. This is in a triple dividend for entropy: less imports of food reduce entropy subject to food imports and habitation.

However, this raises the question of how many inhabitants "should" be accommodated (as food is needed) and are we still correct with respect to the objective function? Since there are system-wide effects which, for instance, concern assuring space of a certain population (existing) and simultaneously reducing industry (income), we must include such concerns and discussion extensions (below). Note that we are looking at less pollution by the overall city and per inhabitant. Minimizing complexity, habitants shall be accommodated, though, which means the fringe has to be extended.

## 8 Planning as a way of addressing objectives: Entropy target and cost minimization

For our planning, yet as a control problem (so far, it still has to be sketched below), the above functions only exhibit policy effects, i.e. we can provide a systematic outline of the use of instruments (subsidy and tax) and effects for which we need planning, but planning along objectives is still missing. Planning has to follow explicit objectives, in our case also stretching from the city centre to the periphery in terms of distance, and then summarized systematically. The occupied area shall change which makes choices regarding instruments relevant, since area use has welfare and entropy implications. If instrument variables are tax (on entropy) and subsidy (farming), different people and their welfare are affected. People are residential and location matters. Moreover, we perhaps need to show how land values and entropy perform in a similar functional mode of distance (after planning), here because land values depict propensity and richness of city zones. Then we need a reference and must show impacts, i.e. we need to show how to minimize economic effects given rise in entropy (the prime target is to improve entropy).

Before planning, objective functions must be clarified, though. As a target we take "entropy improvement" [70], which is the accumulated entropy of the system. Then we claim minimization of economic costs, which apparently lower the creation of welfare from entropy. Vice versa, as economic analysis tells, one can analyse how to minimize entropy given a constraint of maximal allowed social welfare reduction, i.e. a target of reduction gets a shadow price and finally income reduction is a constraint. (Note that for the moment distributional aspects are not tackled, although they can be included. Notice they become important if poor people live in the periphery). Social welfare, as treated here, is consumer and producer surplus from entropy; note that this is a starting point. Welfare is accumulated distance-wise. Technically (as shown below), the integral (sum of location along distance) prevails.

A short note: as we have to work with stock and flow variables, the objective should also show implications if both are affected. In this regard, the value of land (stock) is a cost factor for inhabitants, farmers and industry; stock of land values show scarcity. But only owners receive income by rents, users do not. As we are not planning for stockholders but rather for users of land, these values are exogenous to our objective. Note that the issue of utility and rent is difficult to solve for cities [68]. Rents might be inflicted by shares of land use and scarcity, but that is a story to be followed up. As has been said, for the time being we want to reduce complexity and think that consumer and producer surplus are sufficient for a first analysis. However, there is scope for an extension of spatial economics and entropy if land rents get involved. For now, flow variables are given as integrals (below). They shall suitably show system-wide impacts, while objective function variables are linked through expression of instruments. The idea is to apply indirect planning by taxing entropy emitters and subsidizing mitigation through peri-urban agriculture.

## 9 Extension in scope

### 9.1 Quarrels with respect to residence numbers, need for food and new city appearance

Having stated some system-wide effects within a given frame of the urban fringe and indicated that the system (city) will "naturally" expand, it becomes necessary to complement the analysis with regards to a "new" city. The new city should, in particular, offer a complementary residence expansion into the hinterland or be limited. If the parallel objective is to accept the same number of inhabitants, things become complex. Note that we do not claim the same for industry; we rather regard industry per spatial unit as independent of entropy and soil nutrient

metabolisms which are established primarily by food and recycling of soil nutrients (organic waste). One reason for the special role of industry is that it is under another metabolism, not discussed here. We accept that humans need shelter, while industry does not change. It works at the same rates as in the initial city, although jobs in industry may be floating. Another question regarding the entropy of industry concerns a lesser population density or respectively whether we have a gradient concerning hazard.

In fact, as shown in the first part, a system analysis requires us to define system boundaries and to find definitions of further interactions beyond the current urban fringe. If the number of inhabitants shall be given and the city expansion is subject to a shadow price detection of land prices, there is seemingly no way to avoid expansion. However, if the expansion cannot be unlimited the planner has to set rules for expansion based on the logic of entropy minimization. Given economic, social and physical constraints (sustain livelihoods) prevail. Furthermore, if we concede that initially food has been imported (as opposed to the drawing in Diagram 10), we have to deal with entropy and food imports; importing limited amounts of food can be a policy, as well. This has different implications for entropy. We believe that not all food can be produced within a city. In order to solve the issue, in terms of entropy and its detection, our analysis claims an external target of entropy from food imports and claims a drop of entropy subject to food imports. In this respect, a trade-off can be modelled which ensures that less system entropy is compensated for by more food.

Including food import reduction of mega-cities shall be another major target. It can be set (in modelling, below) by an accumulated target of local food. Apparently, this target is to be found at the system boundary level. It leaves scope for the process of locating the bulk of farming. Also it translates into another entropy reduction target concerning sizes. Any planning becomes an issue of optimal city size. Mathematically the terminal condition of the control problem becomes flexible. This is to say that if the planner has been given a functional constraint regarding the outline of the boundary with respect to the distance, one can proceed to implement a food constraint in the system analysis. For example, the city planner has to contemplate what has been produced beyond the fringe so far. If it has been food, balance improvements within the city have to be adjusted or losses for residence and industry will prevail. Within the given spatial frame which balances the pros and cons, residual food and spatial effects are modelled. This modelling is feasible since we have underlying functional forms, which extend into space beyond the current fringe.

## 9.2 Further assumptions on treating the hinterland or needed city expansion

As we perhaps want a minimal city expansion [4 and 55] planning needs to integrate indications on limits to growth. As has been said, one limit is the expansion of population into the hinterland. Another limit is the derivation of negative flows into the hinterland, as well as understanding limitation as normative for an "ideal city". We need a proportional definition of hinterland functions. What does that mean? Approaching the issue from the definition of an agrarian hinterland as low fragmented and intact networks as well as an organized system of least entropy, "ideal" shares of land use could be defined where the hinterland has a function of mitigation. It allows a definition of shares which are normatively retrievable for targets [67]. The background approaches an average sharing of residence, agriculture and industry which is least fragmented and sustains ecological networks incl. hinterlands. It also includes an accepted entropy which can be observed beyond an urban fringe. Technically it translates into land use shares which offer a definition of the system boundary based on thresholds. But regarding the treatment of technical jargons we must get an idea on the programming method, i.e. control theory, now.

## 10 Planning and policy derivation as control theory (Calculus of variation) approach

### 10.1 Stating the objectives within the accomplished and simplified system outline

We suggest to minimize losses from consumer and producer surplus incurred by entropy reduction (exploitation). Entropy improvement is given as the target of the accumulated entropy of the system "mega-city". This can be stated an upper limit of the externality of the city. We take surpluses as social welfare. For "social" costs in terms of reduced private welfare one can apply private vs. public behaviour (whereas welfare is so far derived from an institutional failure, i.e. the source of externality is referred to as free-riding on externalities; in our case entropy [68]. (Note that negative externality and talk on land use decisions without planning is referred to as missing public aspects in welfare analysis; in other words, we use market solutions for land use as reference.)

We claim that the objective function for our control problem is Eq (7). As has been pointed out, it can be obtained from a producer-consumer-surplus analysis. Then, as Bromley [69] has argued for shape of welfare generation by externality, private utility and income are distance dependent. Thus we claim that closer to the centre the loss may be bigger if instruments are more intensively applied. This is off-set by a lower application in the peri-urban. What is decisive is land sharing. The process of finding corrected system welfare is expressed as integral (7). It includes public goods, policy, needs, reference, deviations, etc. [64]. We portray producer and consumer surplus as dependent on entropy E and land tax t: firstly, without food, i.e. food is not related to entropy. This means, in case of costs, an integral also expresses summation over $\tau$ for total costs:

$$\int_{i^D} [E_i - E_{i,0}]\, a_i\, p_i + [E_h - E_{hi,0}]\, a_h\, p_h]\, \exp\{\varepsilon\tau\}d\tau$$

where additionally

$\tau$ distance

$p$ value added per area unit    (7)

$\varepsilon$ declining preference to periphery

Then, pricing for industry (at location $\tau$) is a function of tax, which gives a behavioural function,

$$[P_i(\tau) - t] = \varepsilon_{b,i,1}[E_i - E_{i,0}] - \varepsilon_{b,i,2}[a_i - a_{i,0}] \tag{8}$$

It says that we do a change partially for land use; i.e. if we tax industry for local entropy with $t(\tau)$: $E_i$ responds by using less land.

For households or inhabitants as land is used by residents, the tax $t(\tau)$ gives a similar equation:

$$[P_h(\tau) - t] = \varepsilon_{b,h,1}[E_h - E_{h,0}] - \varepsilon_{b,h,2}[a_h - a_{h,0}] \tag{9}$$

For agriculture we refer to subsidization $s(\tau)$. The response on policy, i.e. subsidized food, is

$$[P_f(\tau) + s{\cdot}t] = \varepsilon_{b,f,1}[E_f - E_{f,0}] - \varepsilon_{b,f,2}[a_f - a_{f,0}] \tag{10}$$

Now we receive area coding as change and explained by the use of instruments in space. In our case the variables $E_h$, $E_i$ are taxed (system performance: reduced) and $E_a$ is augmented (but has a lower subsidy). The interaction of tax and subsidy, as budget constraint has to be further explained.

The explanation is, we supplement the objective statement by a technically and politically endorsed statement on the financial implications, i.e. the budget variable, as flow variable is endogenous. In other words money obtained is spent. The scheme shall be financially neutral, i.e. the tax receipt minus expenditure is zero. We can state this balance for sites or make it flexible.

$$t(\tau) \left[ a_h\, E_h\, (\tau) + a_i\, E_i\, (\tau) \right] = s(\tau) a_i\, \psi\, C \Leftrightarrow t = C/E\, s \text{ and we presume a fix rate}$$

where additionally                                                                                                    (11)

$\psi$ and C are fixed coefficients per area code

Note that, as assumed in (11) it may be better to work on carbon emission and savings compared to entropy. Yet, per unit we receive a constraint linking taxing and subsidization on the basis of instruments. Formula (11) is an expression of the different land-use systems as impact of policy in terms of emission/mitigation. Some mitigation is financially reached by subsidization and taxing. For any location and in total expenditure as well as expenses for the controlling body, the planning system shall give an interrelated answer. It is even possible to take the system-wise balancing.

$$\int_i^D [(E_i\, a_i + E_h\, a_h)t - s_f\, a_f]\, d\tau = 0 \tag{12}$$

Eq (12) means that we apply financial neutrality and focus on welfare changes imposed by reduced externality of the city (inverse entropy). For the objective and ecological performance of our system (accommodating more agriculture in the peri-urban) neutrality of financial instruments says that government does not spend money on targets; instead the wished target as entropy is reached by cost minimization. We follow the strategy that target and costs of approaching targets are neutral for government. Neutral in finance means there are no government expenditure.

## 10.2 Applying control theory as mathematical tool in planning without city expansion

In this chapter we will briefly demonstrate the power of control theory (calculus of variation by Tu: [61]) for planning and optimization of policy instruments. It works also in regard to land use change in a spatial outline of a city applying a radius as dimension (radius instead of time); in our case it relates to optimization of the instruments differently in space: subsidization and taxing to achieve indirect responses. Three sources for entropy (change) prevail: industry and household (negative) and peri-urban farming (positive). Policy improves the system, with respect to food and recycling. The direct instrument variable is: "subsidy"; the indirect is: "land use". As indicated in Diagrams 10–12, effects depend on land use wedges and distance to city centre which is the agglomeration. As land use sharing (in %, vertical distances) is essential and as is impacting entropy, system performance is addressed. In this regard, a wedge of farming closer to the city centre will decrease entropy (positive, resp. lower emission, negative) and control theory delivers $s(\tau)$ and $t(\tau)$ functions.

Based on the description of our system, which represents an urban-peri-urban-rural interface, a reduced form for control [61] can be derived, enabling optimisation by dynamic

programming.

$$O(\Lambda, E, s, \tau) = \Lambda(D)E_s(D) - \int_{0^D} E_s(\tau)d\tau\} - \int_{0^D} \exp\{\epsilon\tau\}\{E(\tau)[\omega_{t0} - \omega_{t1}E(\tau) - \omega_{t2}s(\tau)]\}d\tau$$
$$+ \int_{0^D} \lambda(\tau)\{\dot{E}(\tau) - \omega_{41}s(\tau) + \Theta_{0s}\cdot\exp\Theta_{1s}\tau\}d\tau \tag{13}$$

For the sake of messaging, we have minimized complexity to a reduced form of the above complex system. In terms of control theory—for a spatial control which has the distance to the city centre as a dimension—Eq (13) links to a Hamiltonian expression [67] in which integrals are dropped

$$H(\Lambda, \lambda, E, s) = \Lambda(D)E(D) -\!\!-\exp\{\epsilon\tau\}E(\tau)[\omega_{t0} - \omega_{t1}E(\tau) - \omega_{t2}s(\tau)] + \lambda(\tau)[\omega_{40}E(\tau)$$
$$- \omega_{41}s(\tau) + \Theta_{0s}\cdot\exp\Theta_{1s}\tau] \tag{14}$$

03Entropy "E" is the stock variable and subsidization "s" the control variable over distance. This simplification is made possible by financial neutrality (above). Derivatives for optimization and optimal planning give 3 criteria to: $\Lambda, \lambda, E, s$

This Hamiltonian expression can be optimized along the criteria:

$$\delta H(.)/\delta E = \Lambda(D) - \exp\{\epsilon\tau\}\omega_{t0} - \omega_{t1}2E(\tau) - \omega_{t2}s(\tau) = -\lambda^{\cdot}(\tau) + \epsilon \cdot \exp\{\epsilon\tau\} \tag{15.1}$$

$$\delta H(.)/\delta s = +\epsilon\cdot\exp\{\epsilon\tau\}[\omega_{t1}E(\tau) - \omega_{t2}] - \omega_{41}\lambda(\tau) = 0 \tag{15.2}$$

$$\delta H(.)/\delta\lambda = -\omega_{40}E(\tau) + \omega_{41}s(\tau)] + \Theta_{0s}\cdot\exp\Theta_{1s}\tau = \dot{E} \tag{15.3}$$

Moreover, a terminal condition for total externality $E_s$, in the simple case of no city expansion, is set by a max. distance D:

$$E_s(D) = \int_{0^D} E(\tau) \, d\tau \tag{16}$$

In a primal solution this is actually the target for the accepted (planned) entropy of the city as a system. Then Eq (15) to (16) offer cost minimization for such a given goal and the result is a regional strategy (differential equation). Finally, the system of optimal subsidization and hence land use (plan) (ref. to 11) is given by a new (exponential) function for land planning which is endogenous. Note that we get it by combining and modifying (2). Technically and mathematically, one can apply mixed differential Eq of (15) for numerical solutions [65]. The needed data and practical computing have to be made available in the explained mathematical way (see above).

## 10.3 Options for applying control theory as a mathematical tool in the extension of a given city framing

Reviewing the limits of previous aspects of planning, we need an extension of analysis for potential city expansion because of more land in peri-urban (chapter 9). It says a certain set-up of a city already prevails and population has to be accommodated. For example, the number of inhabitants, area occupation (relocation or taxing and "voluntary" move) and economic contribution to regional value added become modified. Basically, we also must find novelties. Technically, novelty can be included into a new control, resp. programming problem (calculus of variation at distance), but functions must (can) be changed. Indeed, our approach has worked for the final distance D. (It is radius-oriented and the underlying spatial outlay becomes a wedge in a circle in terms of synergies and modelling). With improvements we can apply (simulate) joint effects (Diagram 13). With regard to framing we see two options: (1) An expansion of radius into the hinterland which accommodates people losing privileges for

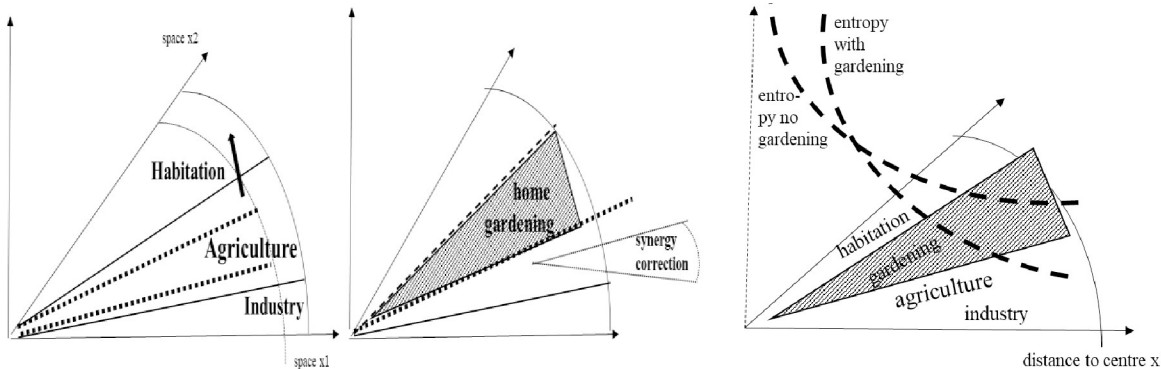

**Diagram 13. Extension and Inclusion of Innovation such as 13.1 Initial.** 13.2 Habitation and Gardening. 13.3 Entropy Performance. Source: own design.

homesteads within a given radius, resp. they get financial incentives to move further out. (2) One can work with technological change which makes entropy effects lower and different. The two cases are briefly sketched towards their modelling background.

In the first case, given current distance D, a further constraint is introduced, summarizing loss of residential space with the current fringe and balancing it with newly to be established habitants within a hinterland. Diagram 13.1 illustrates the case of an increasing periphery. This is actually a very likely case. However, modelling will change, but control theory can go ahead.

In the second case (Diagram 13.2), indeed, several options of technology change can prevail. One probable case is a closer linkage between agriculture and waste management in collecting organic materials including facets, urine, etc. Such innovations reduce external soil nutrient needs and is apparently land saving. This means that novelties come into play of residence (different buildings).

And living at the fringe is possible if there is a closer integration of residence and farming. In terms of modelling we can change entropy generation functions and include, for example, for habitation

$$\dot{E}_h = \varepsilon_{h,1}\, E_{ih} + \varepsilon_{h,2}\, a_h/a_t - \varepsilon_{h,a3}\, [a_h \cdot a_f]/a_t + \Theta_{1,0h} \cdot \exp \Theta_{1h} \cdot \imath \qquad (3')$$

The joint effect is a multiplicative combination of farm area having a diminishing effect on entropy and the residence sector, for instance looking at food from gardening instead of buying food from imported sources. Though it complicates the mathematical outline (above), it enables us to see cross effects of the new type of peri-urban farming and land use, for example, home gardening, etc. Land sharing also changes. We can introduce a new category of residence and home gardening.

In modelling a re-formulation of technology is possible, now in space, which acknowledges synergies in quantification of entropy (Diagram 13.3). Indeed, for detailed analysis, the slope of the novelty (technological and/or building or integration as innovation) as well as threshold for its implementation become relevant, as do costs. Without going into detail, we can further sketch the following in modelling: at new distances technologies must be formulated which are internally solved by the regional optimization of local tenants. Comparable approaches, though for time [70], exist and the argument has to be transferred into spatial modelling. Similar approaches have to be reformulated. This includes thresholds and modified functions using terminal constraints.

A further remark: Since there are always the two objectives of improved entropy vs. careful, external cost minimization (incurred from innovation), the choice of technology is crucial. We regard planning for space and looking for innovation as a joint task. It can be grounded in switches at the threshold [70]. Beside the manoeuvre within technologies for spatial outlay, modelling for optimization requires a different accumulation of space (wedges). So far, space has been additive. In case of synergies, a technical modification is the inclusion of joint effects at different radii. Such effects are multiplicative and, as illustrated, the joint effect allows for additional entropy reduction (as a new wedge). The joint effect offers reduction of the required actual space for residence and farming. Diagram 13.2 illustrates the case by introducing a grey zone and using fictional land augmentation (taking out a new wedge) which enables a co-existence of habitation and farming, such as gardening. A typical case is peri-urban gardening, i.e. locating gardens specifically to residence who buy less food and smallholders in new fringes between residence and agriculture. Earlier, this has been stressed in the problem statement (Chapter 3); now it can be simulated. As a transition zone (grey, Diagram 13.3) it adds less entropy per unit to the spatial summing of the existent city.

In fact, new city segments as part of circle economy become more attractive. Segmenting may be given through interior, coarse road networks, access to special infrastructure, etc. Yet this requires new planning in terms of local governance. However, in the course of modelling the system (by the above functions), we claim a joint effect of angles in the circle economy based on distance in order to get the integrals in space. (For a simple version, three sided spatial units of farming and habitation are added.) Technically and fictionally expressed in Diagram 13.2, a slice is broken down and filled by the innovation for mitigating entropy recycling (as given in our case of gardening and organic matter, slurry, etc.).

Finally, to fit all areas together between space and entropy, factual measures (technologies and policy) as well as images matter. For example, a joint effect can be compost making and "drainage" of slurry as mentioned in the beginning. However, this incurs additional costs and needs optimization of the drainage system. Then it links, in closer contact, homesteads and fields by a new special city structure and organization. Empirically one has to find the right structure and the thresholds for implementation. For the moment we just indicate structures as elements characterized as new grey zones; they are providers of entropy improvement. The job of modelling is to show where this fits into a "city" at minimal costs. Secondly, we just portrayed mathematical and technical aspects of planning and indicating spatial priorities. In fact, in the overall outline, industry is also affected if a certain number of residents need jobs. It must be lodged in the neighbourhood, which again -technically- is a constraint in the optimization, otherwise it contributes to entropy.

## 11 Discussion and conclusions

Firstly, we argued that it is necessary to come up with multi-disciplinary interfaces for peri-urban agriculture and to propose methods that go beyond current disciplinary ones. We see entropy as central in the case of mega-city development. Then we gave qualitative and theoretical arguments for problems in city development. After that we sketched a quantitative method for planning.

Concepts and methods shall contribute to defining increased sustainability in the peri-urban, yet as a special subject of urban planning, and planning based on a programming model enable simulations. We outline the importance of peri-urban areas and rural hinterlands in continuous space and set ground for strategic optimization. Since several strains of thought had to be integrated; they were combined in a common platform designed to understand the overall problem. In particular, we contributed to knowledge for the gathering of

ecological information and planning which have been discussed along entropy. A new way of how to structure land use about entropy effects of mega-cities was built. Then we argued that urban sustainability has to be grounded in common thoughts on generalised systems and has to contribute to system sustainability.

With respect to the above-mentioned problem of finding a continuous change in the space between urban, peri-urban and rural areas, further work should be of interest, such as zoning of newly introduced farming practices and residence modes. Perhaps, if areas are no longer either urban nor rural, new modes of integrated living, for instance in garden city suburbs can be checked. However, we did not argue for strict and direct zoning, but rather view taxes and subsidies as indirect instruments. However, these instruments have to be supported by planning. In a unified research concept on sustainability that integrates economic, ecological, social aspects as well as innovations, the formation of functions conducted by people in a community is key and must be checked in terms of entropy. In particular, specific technological change and people in areas with appropriate technologies need to be addressed, here urban farming is an option. The author pleads for developing applications that help local food business instead of advertising for international trade.

The subject to be investigated further is whether an externally driven growth, which is the pertinent strategy of development in many leading regions, may lead to changes or planning. Within urban areas we need planning and we need to find out whether, alternatively, locals can get more involved in initiatives which compete with outside business for food, in particular local food and recycling. Maintaining the status quo poses a threat. The identification of the scope for functions in recycling of nutrients must be reckoned from the point of location. Hence, one can investigate whether there is a new gradient of land use needed and what the deficits between urban and peri-urban areas are? As qualitatively discussed, we think that making cities more viable depends on entropy concerns. Then, for a quantitative analysis, we presented a modelling approach for regional planning which is meant to help us understand interactions between emission (targets), entropy (aim) and land use (goals). Basically such modelling shall enable simulations. Yet, we believe that received taxing and subsidization will have only mid-term effects, but perhaps stabilize current land use. We hope that, instead of further grabbing land by way of city intensification, our suggested policy helps in maybe sustaining a status-quo for still remaining agricultural land which is used for recycling.

One conclusion is that research for peri-urban land use needs much more detailed deliberations on embedding said land use in concepts such as entropy, energy use, recycling, etc. For that planning and simulations there are devices which have scope. Achieving a systematic inclusion of specially defined land use practices, which fit both city needs and rural areas, enables us to set priorities. Land use is an optional meta-concept which contains options for sustainable cities and perhaps for organic matter composting, recycling, gardening for fresh fruits, new jobs in farming, open space for recreation, etc.

## Author Contributions

**Conceptualization:** Ernst-August Nuppenau.

**Investigation:** Ernst-August Nuppenau.

**Methodology:** Ernst-August Nuppenau.

**Project administration:** Ernst-August Nuppenau.

**Writing – original draft:** Ernst-August Nuppenau.

**Writing – review & editing:** Ernst-August Nuppenau.

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
