## [Decision Letter · Decision Letter 0]

21 Mar 2023

PONE-D-23-05074Contribution of Agriculture and Peri-Urban Land Use to Entropy and Food of Mega-Ci­ties: On Re­cycling of Organics, Sustainabilty and Land Use Planning by Control TheoryPLOS ONE

Dear Dr. Nuppenau,

Thank you for submitting your manuscript to PLOS ONE. After careful consideration, we feel that it has merit but does not fully meet PLOS ONE’s publication criteria as it currently stands. Therefore, we invite you to submit a revised version of the manuscript that addresses the points raised during the review process.

Please consider all comments of all reviewers 

We look forward to receiving your revised manuscript.

Kind regards,

Ahmed Mancy Mosa, Ph.D.

Academic Editor

PLOS ONE

Journal Requirements:

2. Our internal editors have looked over your manuscript and determined that it is within the scope of our Sustainability and the Circular Economy Call for Papers. The Collection will encompass a diverse and interdisciplinary set of submissions related to sustainability and the circular economy, focusing on production models, business plans, and the contribution of global initiatives to increased sustainability in economic, environmental, and social terms. Additional information can be found on our announcement page: Sustainability and the Circular Economy - PLOS Collections . If you would like your manuscript to be considered for this collection, please let us know in your cover letter and we will ensure that your paper is treated as if you were responding to this call. If you would prefer to remove your manuscript from collection consideration, please specify this in the cover letter.

"..."

"....."

Reviewers' comments:

Reviewer's Responses to Questions

**Comments to the Author**

1. Is the manuscript technically sound, and do the data support the conclusions?

Reviewer #1: Partly

Reviewer #2: No

2. Has the statistical analysis been performed appropriately and rigorously? 

Reviewer #1: No

Reviewer #2: I Don't Know

3. Have the authors made all data underlying the findings in their manuscript fully available?

Reviewer #1: Yes

Reviewer #2: No

4. Is the manuscript presented in an intelligible fashion and written in standard English?

Reviewer #1: Yes

Reviewer #2: No

5. Review Comments to the Author

Reviewer #1: Do revise and reduce the length of this manuscript for better comprehension. Include case studies from different region where agriculture is their main economic strength. Discussion along this line will make this manuscript a good one.

Reviewer #2: The abstract lacks a background theoretical basis on which the whole article is grounded. Please revise the abstract to capture some aspects of the conceptual framework of the work and the theory being propounded. As it stands now, it lacks the readings interest that engages readers to want to go further and read the whole article.

The introduction to the study is not clear; so many concepts are discussed at the same time: urban-peri-urban-rural land use systems, ecological-economic-social scales entropy, etc making it difficult to grasp the central theme of the background introduction.

Grammar is very poor and needs to be greatly improved. The whole write-up is saddled with grammatical errors, no coherency in flow of ideas and the ideas are disorganized and disjointed from each other. Because of this, it makes reading of the manuscript difficult and attempt to understand what the write-up is about is fruitless.

All diagrams are difficult to read and perceive; and to link to the written concepts.

Authors need to re-organize their thoughts and put in more effort to express them in writing before re-submitting for review consideration. The topic idea is good, however, the data and writing is poor and do not give enough credit to the topic.

6. PLOS authors have the option to publish the peer review history of their article (what does this mean?). If published, this will include your full peer review and any attached files.

Reviewer #1: No

Reviewer #2: No

---

## [Author Response · Author response to Decision Letter 0]

20 Apr 2023

I slightly changed the title because of reviewers comments.

---

## [Decision Letter · Decision Letter 1]

7 May 2023

PONE-D-23-05074R1Contribution of Peri-Urban Land Use and Agriculture to Entropy and Food of Mega-Cities: On Sustainability, Re­cycling of Organics and Planning by Control TheoryPLOS ONE

Dear Dr. Nuppenau,

Thank you for submitting your manuscript to PLOS ONE. After careful consideration, we feel that it has merit but does not fully meet PLOS ONE’s publication criteria as it currently stands. Therefore, we invite you to submit a revised version of the manuscript that addresses the points raised during the review process.

Please consider all comments

We look forward to receiving your revised manuscript.

Kind regards,

Ahmed Mancy Mosa, Ph.D.

Academic Editor

PLOS ONE

Journal Requirements:

Reviewers' comments:

Reviewer's Responses to Questions

**Comments to the Author**

1. If the authors have adequately addressed your comments raised in a previous round of review and you feel that this manuscript is now acceptable for publication, you may indicate that here to bypass the “Comments to the Author” section, enter your conflict of interest statement in the “Confidential to Editor” section, and submit your "Accept" recommendation.

Reviewer #1: All comments have been addressed

2. Is the manuscript technically sound, and do the data support the conclusions?

Reviewer #1: Yes

3. Has the statistical analysis been performed appropriately and rigorously? 

Reviewer #1: Yes

4. Have the authors made all data underlying the findings in their manuscript fully available?

Reviewer #1: Yes

5. Is the manuscript presented in an intelligible fashion and written in standard English?

Reviewer #1: No

6. Review Comments to the Author

Reviewer #1: Please send for english editing as i found several grammatical errors which reduced the incredibility of this MS.

7. PLOS authors have the option to publish the peer review history of their article (what does this mean?). If published, this will include your full peer review and any attached files.

Reviewer #1: No

---

## [Author Response · Author response to Decision Letter 1]

29 Jun 2023

I followed the request for English and grammar corrects as mailed 08.05.2023.

---

## [Editor Report · Decision Letter 2]

15 Aug 2023

Contribution of Peri-Urban Land Use and Agriculture to Entropy and Food of Mega-Cities: On Sustainability, Planning by Control Theory and Re­cycling of Organics

PONE-D-23-05074R2

Dear Dr. Nuppenau,

We’re pleased to inform you that your manuscript has been judged scientifically suitable for publication and will be formally accepted for publication once it meets all outstanding technical requirements.

Kind regards,

Ahmed Mancy Mosa, Ph.D.

Academic Editor

PLOS ONE
---

## [Editor Report · Acceptance letter]

16 Aug 2023

PONE-D-23-05074R2 

Contribution of Peri-Urban Land Use and Agriculture to Entropy and Food of Mega-Cities: On Sustainability, Planning by Control Theory and Re­cycling of Organics 

Dear Dr. Nuppenau:

I'm pleased to inform you that your manuscript has been deemed suitable for publication in PLOS ONE. Congratulations! Your manuscript is now with our production department. 

Kind regards, 

on behalf of

Dr. Ahmed Mancy Mosa 

Academic Editor

PLOS ONE